The first reported ceratopsid dinosaur from eastern North America (Owl Creek Formation, Upper Cretaceous, Mississippi, USA)

http://orcid.org/0000-0002-6930-2002 Farke Andrew A. 1 afarke@webb.org
http://orcid.org/0000-0002-2173-8544 Phillips George E. 2
1 Raymond M. Alf Museum of Paleontology , Claremont, CA , USA
2 Mississippi Museum of Natural Science, Mississippi Department of Wildlife, Fisheries, and Parks , Jackson, MS , USA
Sues Hans-Dieter
Electronic publication date: 2017 May 23
Publication date: 2017
Volume: 5
Electronic Location ID: e3342
Received 2017 Jan 25; Accepted 2017 Apr 21
Copyright: © 2017 Farke and Phillips
Copyright year: 2017
Copyright holder: Farke et al.
License: This is an open access article distributed under the terms of the Creative Commons Attribution License, which permits unrestricted use, distribution, reproduction and adaptation in any medium and for any purpose provided that it is properly attributed. For attribution, the original author(s), title, publication source (PeerJ) and either DOI or URL of the article must be cited.
License URL: https://creativecommons.org/licenses/by/4.0/

Keywords: Ceratopsia, Biogeography, Laramidia, Appalachia, Ceratopsidae, Dinosauria, Owl Creek Formation, Cretaceous, Dinosaur, Western Interior Seaway

Funding: The authors received no funding for this work.

==============================
Ceratopsids (“horned dinosaurs”) are known from western North America and Asia, a distribution reflecting an inferred subaerial link between the two landmasses during the Late Cretaceous. However, this clade was previously unknown from eastern North America, presumably due to limited outcrop of the appropriate age and depositional environment as well as the separation of eastern and western North America by the Western Interior Seaway during much of the Late Cretaceous. A dentary tooth from the Owl Creek Formation (late Maastrichtian) of Union County, Mississippi, represents the first reported occurrence of Ceratopsidae from eastern North America. This tooth shows a combination of features typical of Ceratopsidae, including a double root and a prominent, blade-like carina. Based on the age of the fossil, we hypothesize that it is consistent with a dispersal of ceratopsids into eastern North America during the very latest Cretaceous, presumably after the two halves of North America were reunited following the retreat of the Western Interior Seaway.

Introduction

The Western Interior Seaway split North America during much of the Late Cretaceous, which in turn may have driven terrestrial faunal differences between eastern and western North America (Appalachia and Laramidia, respectively). Non-avian dinosaur fossils from the Late Cretaceous of Appalachia are, with a few notable exceptions, largely fragmentary and indicative of a fauna including theropods (ornithomimosaurs and tyrannosauroids), nodosaurids, hadrosauroids, and potentially leptoceratopsids (Schwimmer, 1997; Weishampel et al., 2004; Longrich, 2016; Prieto-Márquez, Erickson & Ebersole, 2016a). The hadrosauroids and tyrannosauroids in particular have been suggested as representing clades distinct from their relatives in western North America (Longrich, 2016). This is further supported by the notable absence of ceratopsid dinosaurs, which are abundant in Laramidia, from the published fossil record of Appalachia. Faunal differences between Laramidia and Appalachia presumably were reduced when the two land masses rejoined following the retreat of the interior seaway during the late Maastrichtian (if they were indeed rejoined; see Slattery et al., 2015 for a discussion of this issue). Yet late Maastrichtian fossils of terrestrial origin are virtually unknown from eastern North America, so there is little evidence to test this hypothesis.

Here, we report the first definitive ceratopsid specimen from eastern North America, a tooth recovered from the Maastrichtian Owl Creek Formation of Union County, Mississippi. The fossil, collected by the second writer (G. E. Phillips) in July 2016, suggests a dispersal of ceratopsids into eastern North America following the regression of the Western Interior Seaway.

Geologic Setting

Occurrence

The tooth described here (MMNS VP-7969) was collected in loose association with the Upper Cretaceous marine Owl Creek Formation (and other units) in northeast Mississippi (Fig. 1). More precisely, it was found out of context in the active fluviatile lag of a modern stream, albeit probably in close proximity to its presumed stratigraphic origins. The pebbly, fossiliferous stream lag contains Pleistocene terrestrial-alluvial, Paleocene marine, and Cretaceous marine fossil float originating from the channel floor and (to a limited extent) the walls. The Paleocene is represented in the area by the Clayton Formation (Fig. 2), the nearest outcrop (preserving the base of the formation) of which is ∼4.3 km upstream (and up-section) from the tooth collection point. Fossil float originating from the Clayton Formation has been limited to fragments of the Paleocene index gastropod Kapalmerella mortoni (Conrad, 1830). Based on the extent of channel length explored thus far, Quaternary alluvium, slumping, vegetation, and water level conceal the underlying Owl Creek Formation (Upper Cretaceous) rather thoroughly, making direct access to the Owl Creek beds very difficult. Although rarely exposed in the stream, these beds crop out intermittently along the channel length between the base of the Clayton and the tooth recovery point. The tooth was retrieved from the stream float within a few meters of the contact between the Owl Creek Formation and the subjacent Chiwapa Sandstone Member of the Ripley Formation at Mississippi Museum of Natural Science (MMNS) locality MS.73.001b (Fig. 1).

Figure 1 Geologic map of Maastrichtian deposits in northeast Mississippi.

The area of interest includes the noteworthy type localities of the Coon Creek Formation (latest Campanian–early Maastrichtian) and Owl Creek Formation (late Maastrichtian). Base map composed by the Mississippi Office of Geology in 2010, from data in Bicker (1969).

Figure 2 Stratigraphic chart of Maastrichtian deposits in northeast Mississippi.

Basic chart chronostratigraphy and most of the biostratigraphic columns were produced using TS (TimeScale) Creator (Ogg & Lugowski, 2012). All ages are standardized to the Geologic Time Scale 2016 and the Concise Geologic Time Scale compilation of the International Commission on Stratigraphy and its Subcommission on Stratigraphic Information. The stratigraphic data used in TS Creator is based on numerous events borrowed from many global and regional reference sections and integrated time scales. The Gulf Coastal Plain (GCP) ammonite zones and their correlative ages are based primarily on Cobban (1974), Cobban & Kennedy (1991a, 1991b, 1995), Kennedy & Cobban (1993), Landman, Johnson & Edwards (2004) and Larina et al. (2016). The relationship of GCP to WIS ammonite zones as presented here should be considered provisional. The position of the stage and substage boundaries is based, in part, on the work of Sohl & Koch (1986). The informal units “Nixon beds,” “Troy beds,” and “transitional clay” were introduced by Phillips (2010), Swann & Dew (2008, 2009), and Sohl (1960), respectively. The Coon Creek and correlative beds are time transgressive, the Campanian–Maastrichtian boundary being located higher in the section in the northern part of the outcrop belt (Tennessee). A major unconformity is recognized at the base of the Chiwapa Sandstone, separating it from the remainder of the subjacent Ripley Formation. Contrary to the age of the sub-Chiwapa Ripley given here (early Maastrichtian), foraminiferal zonation established for the Gulf Coast by Mancini et al. (1995) and Puckett (2005) defines the Campanian–Maastrichtian boundary as coincident with the transgressive surface marking the base of the Chiwapa Sandstone, thus making the lower Ripley beds Campanian. The dashed vertical arrow represents the uncertainty of the exact stratigraphic position for the ceratopsid tooth within the Owl Creek Formation.

Both the Cretaceous and Paleocene units cropping out in the channel contain marine vertebrate fossils, although vertebrate fossils are considerably more common in the former than in the latter. Cretaceous deposits in the area have previously produced dinosaur fossils, and the Paleocene occasionally contains reworked Upper Cretaceous fossils. Based on observations of several short-lived, partial exposures in the greater vicinity (e.g., MMNS locality MS.73.030), a persistent phosphatic fossil assemblage occurs in the uppermost part of the Owl Creek Formation. This assemblage consists largely of a shell bed of locally common, dark, well-lithified phosphatic mollusk and decapod steinkerns along with less frequently occurring fragments of marine vertebrates—most of which are characteristically Maastrichtian (Fig. 3; Table 1; Baird, 1986; Phillips, Nyborg & Vega, 2014; Martínez-Díaz et al., 2016). The upper Owl Creek steinkern assemblage is conspicuously populated by baculitid and scaphitid ammonites not seen elsewhere in the local Maastrichtian section. These same ammonites are common in the stream float that yielded the ceratopsian tooth. The Chiwapa Sandstone is very fossiliferous, as is the basal Owl Creek Formation. However, the suite of Cretaceous fossils in the float is generally inconsistent with the assemblage contained in either of these intervals. The Chiwapa contains crystalline calcite pseudomorphs of mollusk shells, none of which are scaphitid or baculitid ammonites. Also, the highly lithified Chiwapa Sandstone does not surrender fossils to the stream bed in one piece—shark teeth, bones, and even shells shatter as soon as they begin weathering from the surface of the rocky exposure. Where the ceratopsian tooth was recovered, the basal Owl Creek is exposed and deeply weathered and contains mollusk steinkerns; however, it also lacks the kinds of ammonites consistent with the stream float. Of all the sourceable constituents of the modern stream lag, the ceratopsian tooth is most consistent with the average size, specific gravity, and color of the phosphatic fossils and pebbles that populate the upper part of the Owl Creek Formation.

Figure 3 Marine macrofossils collected in loose association with ceratopsian tooth (from Table 1), most consistent with a Maastrichtian age.

(A) Striaticostatum cf. S. sparsum Sohl, MMNS IP-8648; (B) Liopistha protexta (Conrad), MMNS IP-6116; (C) Discoscaphites iris (Conrad), microconch, MMNS IP-8646; (D) Costacopluma grayi Feldmann & Portell, larger Maastrichtian variety (Martínez-Díaz et al., 2016), MMNS IP-8647 (distinct from the smaller Danian variety); (E) Discoscaphites iris (Conrad), macroconch, MMNS IP-494; (F) Cretalamna appendiculata (Agassiz), variant of a lower posterior tooth, MMNS VP-8041; (G) Branchiocarcinus flectus (Rathbun), MMNS IP-6115.3; (H) Mosasaurus hoffmani Mantell, MMNS VP-6803; and (I) Peritresius ornatus (Leidy), costal carapace fragment, MMNS VP-4407.

Table 1 Partial faunal list produced from upper Cretaceous marine fossils collected in loose association with MMNS VP-7969.

The mollusks were previously established as characteristic of the late Maastrichtian Owl Creek Formation at the type locality, Tippah County, as well as historic outcrops in the vicinity of the ceratopsian locality, Union County (Sohl & Koch, 1983). Many of the other listed species have also been previously reported as distinguishing Maastrichtian marine deposits of the Eastern United States (Baird, 1986; Phillips, Nyborg & Vega, 2014; Martínez-Díaz et al., 2016). Selected specimens are illustrated in Fig. 3.

Mollusca	
 Bivalvia	
  Cucullaea capax Conrad, 1858	
  Tenuipteria argentea (Conrad, 1858)	
  Pinna cf. P. laquata Conrad, 1858	
  Exogyra costata Say, 1820	
  Pycnodonte vesicularis Lamarck, 1806*	
  Pterotrigonia cf. P. eufalensis (Gabb, 1860)	
  Pterotrigonia sp.	
  Crassatella sp.	
  Linearia cf. L. metastriata Conrad, 1860	
  Eufistulana ripleyana (Stephenson, 1941)	
  Liopistha protexta (Conrad, 1853)	
 Gastropoda	
  Turritella sp(p).	
  Striaticostatum cf. S. sparsum Sohl, 1964*	
 Cephalopoda	
  Discoscaphites iris (Conrad, 1858)	
  Trachyscaphites sp.	
  Eubaculites carinatus (Morton, 1834)	
Crustacea	
 Decapoda	
  Branchiocarcinus flectus (Rathbun, 1926)	
  Costacopluma grayi Feldmann & Portell, 2007	
  Palaeoxanthopsis libertiensis (Bishop, 1986)	
Vertebrata	
 Chimaeriformes	
  Ischyodus sp.	
 Selachii	
  Cretalamna appendiculata (Agassiz, 1843)	
  Squalicorax pristodontus (Agassiz, 1843)	
 Testudines	
  Peritresius ornatus (Leidy, 1856)	
 Squamata	
  Mosasaurus hoffmani Mantell, 1829	
Note:

* Mollusks represented by original calcitic shell. Remaining macroinvertebrates are largely internal molds.

The Owl Creek Formation

The Owl Creek Formation crops out in portions of several states within the former Mississippi Embayment—Missouri, Illinois, Tennessee, and Mississippi (Fig. 1). Local thickness of the Owl Creek Formation is about 12 m, and it is rich in Maastrichtian neritic marine fossils (Stephenson, 1955; Sohl, 1960; Sohl & Koch, 1983, 1986). The Owl Creek Formation in northeast Mississippi is composed of glauconitic, variably micaceous, fine-grained beds ranging from sandy clay to clayey sand that become increasingly calcareous to the south where the mostly siliciclastic facies of Tippah and Union counties (including MMNS locality MS.73.001b) grade into the bedded marls and “dirty chalk” of the Prairie Bluff Formation (Stephenson & Monroe, 1940; Sohl, 1960). Thus, terrigenous input in this part of the outcrop belt decreases toward the more pelagic waters of the gulfward shelf. The Owl Creek sediments on the opposite side of the embayment in Missouri and at the head of the embayment in Illinois are texturally and compositionally similar. Likewise, the formation becomes decreasingly calcareous, and then entirely terrigenous, moving northward into the head of the embayment and nearer to the McNairy delta system.

In the first grand interpretation of Upper Cretaceous sedimentation in the Mississippi Embayment, the depositional sequence in the embayment proper was revealed to consist of sediments mineralogically derived from the Appalachian Plateaus and Blue Ridge Mountains (Pryor, 1960). In that study, the Owl Creek Formation was described as an inner prodelta facies of the McNairy Delta complex, although deposited on top of, and partially reworked from, the lower Maastrichtian McNairy Formation during the very last Cretaceous marine transgression into the embayment. In a sequence stratigraphic model, the lower contact of the Owl Creek with the McNairy Sand or Chiwapa Member of the Ripley Formation represents a transgressive surface. Subsequent beds in the Owl Creek would thus represent sediments associated with a transgressive systems tract followed by progradational beds of a highstand systems tract (Mancini et al., 1995).

A palynomorph assemblage from the Owl Creek Formation across the embayment in Missouri suggests an inner neritic marine environment with high terrestrial input (Eifert, 2009). Angiosperms (Betulaceae, Juglandaceae, Oleaceae, Fagaceae, and Nyssaceae) dominate the assemblage, followed by palm (Areaceae) and cycads (Cycadaceae). A foraminiferal suite from the same samples indicates a hypersaline marsh, and a low-diversity/low-abundance dinoflagellate assemblage is inconsistent with a highstand systems tract (Mancini et al., 1995; Eifert, 2009).

Taphonomy

The discovery of dinosaur remains in marine environments occurs infrequently and typically consists of isolated elements or, more rarely, larger skeletal portions (e.g., partial limb or vertebral associations) shed from a bloat-and-float carcass (Schäfer, 1972; Schwimmer, 1997). In this scenario, the buoyant carcasses of coastal dinosaurs, particularly those originating in riparian habitats of tide-dominated estuaries and deltas, are carried to sea by seasonal or episodic freshets and tides. Dinosaur remains from more distal shelf deposits, particularly the more complete skeletal associations, may result from transport enhanced by maritime storms, such as tropical cyclones. Dinosaur fossils in marine sediments seem to be more commonly encountered, and possess greater taxonomic diversity, as fragmentary yet identifiable bones and teeth from nearshore lag deposits (Schwimmer, 1997).

In addition to being the first dinosaur tooth documented from the Owl Creek Formation, the ceratopsian tooth is the first terrestrial macrofossil ever reported from this unit—much-studied previously for its marine macroinvertebrate content. Although characteristically rich in neritic fossils, the aforementioned terrigenous microfossils suggest a not too distant shoreline (Eifert, 2009). Thus, the occurrence in the Owl Creek of a dinosaur fossil, although rare, is not entirely unexpected.

Still, the Mississippi tooth is, literally, one of only a handful of North American ceratopsian fossils from a marine context. Compared to other types of dinosaurs, hadrosaur bones and teeth are the most common dinosaur fossils from Campanian and Maastrichtian marine sediments (Schwimmer, 1997). A possible explanation for the scarcity of ceratopsian remains versus that of other dinosaur taxa recovered from marine deposits may lie in habitat preferences. A summary of generalized ceratopsian lithofacies associations suggests an affinity for “lacustrine, alluvial, and coastal plain” habitats, at least among Ceratopsidae (Eberth, 2010). Alluvial wetland ecosystems can be separated into riparian (channel margin) and more distal floodplain habitats—clast size decreasing with increasing distance from the channel. A study of alluvial wetland lithofacies in the upper Maastrichtian Hell Creek Formation documents a greater proportional contribution of Triceratops remains (out of seven dinosaur families) to floodplain (muddy) over fluviatile (sandy) deposits. The hadrosaur Edmontosaurus is found with greater frequency in the latter (Lyson & Longrich, 2011). If rivers are the principal conveyor of bloat-and-float dinosaur carcasses to the marine realm, then a possible preference among coastal plain ceratopsids for habitats outside of riparian zones may explain their paucity in marine sediments.

The tooth described here exhibits mechanical abrasion (see Description) ostensibly due to fluviatile transport since its exhumation. Thus, a relatively uneroded condition is presumed for the specimen prior to burial. Not knowing the exact stratigraphic origin of the specimen, or whether it fell loose from an as yet undiscovered partial dentary or was buried in isolation, precludes any further speculation as to its postmortem journey and exactly when it entered the Owl Creek depositional system. Nonetheless, based on the locality’s close proximity to the eastern side of the Mississippi Embayment at the time as well as its nearshore sedimentological context (Figs. 1 and 4), we consider it most parsimonious that the tooth originated from an animal in that region, rather than a carcass that had floated from the direction of Laramidia.

Figure 4 Paleogeographic maps of two key geochronologic intervals in the uppermost Cretaceous of North America.

(A) Late Campanian and (B) late Maastrichtian time slices are depicted with southern Laramidia ceratopsid localities on the appropriate time interval map. Ceratopsid occurrences and their associated ages are taken from numerous references (Lehman, 1996; Sullivan, Boere & Lucas, 2005; Loewen et al., 2010; Sampson et al., 2010, 2013; Sullivan & Lucas, 2010; Porras-Múzquiz & Lehman, 2011; Wick & Lehman, 2013; Rivera-Sylva, Hedrick & Dodson, 2016; Lehman, Wick & Barnes, 2016). Arrows designate late Maastrichtian dispersal of ceratopsians, in this interpretation, along an emerging southern route formed by a northerly retreating seaway. We note, however, that the exact placement of any subaerial connection is uncertain (Berry, in press; Boyd & Lillegraven, 2011; Slattery et al., 2015). Although the exact identity of the Mississippi tooth is unknown, we have illustrated only chasmosaurine silhouettes on this part of the figure because no centrosaurines are known from North America during the late Maastrichtian. This Mississippi Embayment is labeled as “Miss. Emb.”. Maps are part of the Key Time Slices of North America series, © 2013 Colorado Plateau Geosystems, Inc., and used with their kind permission by licensed agreement. Silhouettes are by Raven Amos (chasmosaurine) and Lukas Panzarin (centrosaurine, from Sampson et al., 2013), via http://www.phylopic.org.

Age

The Owl Creek Formation lies entirely within the upper Maastrichtian (Fig. 2), according to published ammonite stratigraphy (Larina et al., 2016) and non-cephalopod mollusk assemblage zonation (Sohl & Koch, 1986). Planktonic foraminiferan zonation is consistent with the deposits being at least partly (or mostly) within the upper Maastrichtian (Puckett, 2005), although these are likely less reliable than ammonites or dinoflagellates for identifying that lithostratigraphic interval (Larina et al., 2016). Owl Creek dinocyst composition immediately below the K–Pg boundary on the opposite side of the Mississippi Embayment in Missouri supports a latest Maastrichtian age for the uppermost part of the formation (Oboh-Ikuenobe et al., 2012). Finally, at the head of the embayment in southern Illinois, 40K/40Ar dating of pelletal glauconite in the uppermost Owl Creek Formation yielded an age of 65.7 ± 1.4 Ma (Reed et al., 1977). As indicated above, the exact placement of the tooth within the Owl Creek is uncertain, but associated fossils suggest that it is from considerably closer to the K–Pg boundary (top) than it is to the base of the unit. According to Matt Garb of Brooklyn College (M. Garb, 2016, personal communication), scaphitid ammonite steinkerns in the fossil float accompanying the ceratopsian tooth are almost entirely dominated by Discoscaphites iris (Conrad, 1858; Figs. 3C and 3E), which equates to the uppermost portion of calcareous nannofossil zone CC 26 of Perch-Nielsen (1985) within the latest Maastrichtian (Fig. 2). Thus, we posit that the ceratopsian tooth described here dates to the late Maastrichtian.

Reworking is always a consideration with condensed, phosphatic pebble beds. To date, suspected anachronistic fossils have not been detected at any interval within the Owl Creek Formation. Considering the exceptional condition of the tooth, and that it was collected from modern stream lag below a small waterfall produced by a resistant calcareous sandstone ledge (Ripley Formation, Chiwapa Member), prior to which it had traveled at least several meters across the irregular surface of the exposed sandstone, reworking from a notably older Cretaceous interval prior to entombment in the Owl Creek sediments is highly unlikely.

Methods

In order to illustrate the details of MMNS VP-7969 at high resolution, stacked images were produced with a Visionary Digital Passport system (Dun, Inc., Chesapeake, VA, USA). The stacking device was interfaced with a Canon EOS 6D camera (Canon, Inc., Tokyo, Japan) with attached 50 mm macro lens and a 1.4x Tamron extension, at a magnification setting of 1:2. Images were processed within Helicon Focus 5.3 (Helicon Soft Ltd., Kharkiv, Ukraine).

To produce a three-dimensional digital model for archival and illustration purposes, MMNS VP-7969 was digitized using a NextEngine 3D Scanner Ultra 3D with MultiDrive (NextEngine, Inc., Santa Monica, CA, USA). The initial scans were acquired and processed in ScanStudio PRO 2.0.2 (ShapeTools LLC and NextEngine, Inc., Santa Monica, CA, USA). Data were collected in several passes, with all set for the maximum resolution on the scanner (6,300 points/mm2), using macro mode, and assuming a dark target object. The first pass included six scans taken around the long (apico-basal) axis of the tooth. The second pass included three scans bracketing the apical view of the tooth, and the third pass included three scans bracketing the basal view of the tooth. A final scan captured a portion of the tooth in distal view. The scans were aligned using both manual and automatic alignment, and then fused into a single watertight mesh using the “mesh reconstruction” fuse method (high resolution mesh fitting, and relax fitting selected as an option). This mesh was downsampled to reduce file size, creating a final mesh of 83,312 vertices and 166,620 faces. The file was exported in stereolithography (STL) format and is archived at MorphoSource (http://www.morphosource.org/Detail/SpecimenDetail/Show/specimen_id/4475).

Measurements were taken from the original specimen using digital calipers, to the nearest 0.1 mm. Comparison with measurements taken from the digital model showed the latter to be consistent with the physical specimen to between 0.5% and 2.5%.

All fossils figured and described here are accessioned at the MMNS. The tooth was molded in silicone rubber, and a limited number of plastic resin casts are available to research institutions by placing requests with the MMNS.

Systematic Paleontology

Dinosauria Owen, 1842

Ornithischia Seeley, 1887

Ceratopsia Marsh, 1890

Ceratopsoidea Hay, 1902

Ceratopsidae Marsh, 1888

Ceratopsidae indet.

Referred material: MMNS VP-7969, an isolated right dentary tooth, Fig. 5.

Figure 5 Right dentary tooth of ceratopsid dinosaur, MMNS VP-7969.

Digital renderings and photographs in (A, B) mesial (posterior); (C, D) lingual (medial); (E, F) distal (anterior); (G, H) apical (dorsal); (I, J) labial (lateral); (K, L) root (ventral) views. Scale bar equals 10 mm. Directional abbreviations: api, apical; dist, distal; mes, mesial; lab, labial; ling, lingual.

Locality and horizon: MMNS locality MS.73.001b, Union County, Mississippi, United States of America (Fig. 1); Owl Creek Formation (late Maastrichtian). Precise locality data are on file at MMNS and are available to qualified investigators upon request.

Description: For simplicity, the following description presumes that the tooth is from the right dentary. This is based on the sharply protruding primary ridge, characteristic of dentary teeth in ceratopsids and contrasting with the relatively subdued primary ridge in maxillary teeth. Once oriented as a dentary tooth, the offset of the primary ridge must be in the mesial direction, and the tooth is thus from the right side (Mallon & Anderson, 2014). Terminology follows that illustrated by Tanoue, You & Dodson (2009: Fig. 2).

MMNS VP-7969 preserves both the crown and the root of the tooth (Fig. 5). Portions of the crown were slightly chipped, and the extreme ends of the roots were broken off prior to discovery. Due to dark and consistent coloration across the surface of the tooth, it is not possible to describe enamel distribution with any confidence.

The crown as preserved is taller (18.9 mm) than wide (15.8 mm) in lingual view (Figs. 5C and 5D). A slight peak at the mesial and distal edges, where the root intersects with the carinae, produces a rhomboid profile. A prominent primary ridge divides the tooth crown into a smaller mesial lobe and a larger distal lobe (Fig. 5G). Toward the base of the crown, the ridge has a slight mesial curvature (Figs. 5C and 5D). In mesial and distal views, the primary ridge is strongly arched, and a slight inflection marks the point where the ridge and the cingulum/root connect (Figs. 5A, 5B, 5E and 5F). The primary ridge is fin-like and strongly compressed mesiodistally. The lingual edge of the ridge bears very fine and imbricating crenulations. A single, very poorly defined secondary ridge occurs at the mesial edge of the mesial lobe (Fig. 5C); otherwise, secondary ridges are completely absent. No unambiguous denticles appear on the tooth, either. A distinct cingulum separates the crown from the root on the tooth’s lingual surface (Figs. 5E and 5G). As preserved, the maximum apico-basal length of the entire tooth in lingual view is 26.8 mm.

In labial view, the crown and root are not distinctly separated (Figs. 5I and 5J). The labial surface is gently arched mesiodistally, with at least seven faint plications along the surface of the tooth oriented apico-basally. A flat, approximately quadrangular wear surface marks the apical end of the tooth in this view. A handful of minor scratches mark this area, although the lack of consistent orientation suggests that they are taphonomic in origin rather than representing microwear. Assuming a standard tooth orientation for a ceratopsid, the wear facet was at least subvertical. As preserved, the maximum apico-basal length of the entire tooth in labial view is 28.4 and the maximum width is 16.8 mm.

The root is bipartite, with the two halves having a maximum span of 22.2 mm. The labial root is more robust and longer than the lingual root (Fig. 5E). A v-shaped resorption groove marks the basal surface of the root (Figs. 5K and 5L).

Discussion

Referral to Ceratopsidae

The prominent primary ridge and split root of MMNS VP-7969 definitively distinguish it from teeth belonging to other ornithischian dinosaurs present in North America during the Late Cretaceous, such as hadrosaurs, ankylosaurus, pachycephalosaurs, and basal ornithopods, all of which lack these features. This gross morphology, thus, is most consistent with referral to Ceratopsidae. However, to avoid the hazards of “overidentification,” we here examine the phylogenetic distribution of notable apomorphies in MMNS VP-7969 to arrive at the most conservative identification possible. This is particularly important in light of teeth described for Turanoceratops, a non-ceratopsid ceratopsoid from Uzbekistan that also displays some apomorphies historically recognized only in ceratopsids (Sues & Averianov, 2009; Farke et al., 2009). The subject is further complicated by variation across the tooth row in ceratopsids; teeth at the very mesial or distal end differ from those in the middle in the development of some features (Hatcher, Marsh & Lull, 1907).

Split tooth root

This feature is noted in Turanoceratops tardabilis (Nessov, Kaznyshkina & Cherepanov, 1989; Sues & Averianov, 2009) and all ceratopsids for which the relevant tooth anatomy is preserved, but does not occur in other ceratopsians, nor in other ornithischians as a whole.

Absence of secondary ridges on tooth crown

Secondary ridges paralleling the median carina (primary ridge) are common in teeth of non-ceratopsid neoceratopsians (Tanoue, You & Dodson, 2009), and also occur variably in Turanoceratops (Sues & Averianov, 2009) as well as in Zuniceratops christopheri (A. Farke, 2016, personal observation; AZMNH P2224, AZMNH P3600). Due to their variable occurrence in T. tardabilis, the near absence of these ridges in MMNS VP-7969 can only restrict a tooth to Ceratopsoidea.

Projecting, blade-like primary ridge on dentary teeth

The primary ridge projects strongly from the body of the tooth in MMNS VP-7969 and all ceratopsids, but is far more subdued in dentary teeth of T. tardabilis (Sues & Averianov, 2009: Figs. 2E and 2F) and Z. christopheri (A. Farke, 2016, personal observation; AZMNH P3600). Most notably, in the known Turanoceratops specimens (as well as non-ceratopsoid neoceratopsians such as Protoceratops), the carina is smoothly continuous with the root in mesial and distal views. By contrast, the carina is arched away from the main body of the tooth in MMNS VP-7969 and many ceratopsid dentary teeth (but not all, particularly from those at the extreme ends of the rows). Our observations suggest that the morphology is only found in Ceratopsidae.

In total, the anatomy of MMNS VP-7969 identifies it as a tooth from a ceratopsid dinosaur. At present, a more constrained identification is not possible due to the general similarities in teeth across ceratopsid clades (Mallon & Anderson, 2014). However, only chasmosaurines are known in North America during the late Maastrichtian, so the silhouettes in Fig. 4 are illustrated as such.

Biogeographic and paleogeographic implications

The tooth described here (MMNS VP-7969) represents the first reported occurrence of Ceratopsidae from eastern North America (Appalachia). Previous reports of ceratopsians from Appalachia have been from non-ceratopsid neoceratopsians, including isolated teeth from the Aptian-aged Arundel Formation of Maryland and a potential leptoceratopsid from the Campanian-aged Tar Heel Formation of North Carolina (Chinnery et al., 1998; Chinnery-Allgeier & Kirkland, 2010; Longrich, 2016). The dispersal route of these earlier ceratopsians into Appalachia is uncertain, and the overall evidence supports a lengthy geographic separation of Appalachia from Laramidia during the Late Cretaceous (late Cenomanian to latest Maastrichtian, ∼95–66 Ma, Slattery et al., 2015). Although there is some limited biogeographical evidence for occasional connections between Europe and Appalachia during the Late Cretaceous (summarized in Csiki-Sava et al., 2015), no ceratopsids are known from Europe. So, a European origin for the animal associated with the Mississippi tooth is highly unlikely.

We thus hypothesize that the occurrence of a ceratopsid in Mississippi represents a dispersal event from western North America into eastern North America. Significantly, this is the first time that a representative of this previously Laramidian dinosaur clade has been identified from Appalachia. This provides strong biogeographic evidence for a physical connection between eastern and western North America during the late Maastrichtian (Fig. 4).

Because many regions of the former Western Interior Seaway do not have the relevant strata preserved or accessible, the seaway’s extent during the terminal Maastrichtian has been debated (summarized in Berry, in press; Boyd & Lillegraven, 2011; Slattery et al., 2015 and references therein). For instance, ammonite distribution suggests a marine connection from the Gulf of Mexico northward to South Dakota (but not continuous with marine environments around present-day Greenland) up until the Hoploscaphites nebrascensis biozone during part of the late Maastrichtian (Kennedy et al., 1998). In turn, the shared occurrence of the plant Cissites panduratus between Laramidia and Appalachia during the late Maastrichtian supports a subaerial connection between the two land masses during this time, too (Berry, in press). The ceratopsid tooth in Mississippi provides additional evidence consistent with this scenario.

Eastern dinosaurs

Non-avian dinosaurs from Cretaceous deposits in the eastern US have been well publicized (Weishampel & Young, 1996; Schwimmer, 1997). Although few discoveries are complete enough for comprehensive description and precise taxonomic assignment, recent notable exceptions include a tyrannosauroid and hadrosaurid from Alabama (Carr, Williamson & Schwimmer, 2005; Prieto-Márquez, Erickson & Ebersole, 2016a, 2016b). Cretaceous dinosaur finds from eastern North America are not rare, but they are infrequent. Since Cretaceous dinosaur remains were first reported on the east coast in the 1850s, numerous specimens representing several groups, both ornithischian and theropod, have been reported from Mississippi to New Jersey. Most of this material consists of isolated and often fragmentary elements, like the ceratopsian tooth reported herein. Collectively, however, the scattered discoveries across the Gulf and Atlantic Coastal Plain reveal an eastern North American Cretaceous dinosaur bestiary that included six major dinosaur clades. To date, these include hadrosauroids (Langston, 1960; Prieto-Márquez, Weishampel & Horner, 2006; Prieto-Márquez, Erickson & Ebersole, 2016a), ankylosaurians (Langston, 1960; Weishampel & Young, 1996; Stanford, Weishampel & Deleon, 2011), tyrannosauroids (Baird & Horner, 1979; Schwimmer et al., 1993; Carpenter et al., 1997; Carr, Williamson & Schwimmer, 2005), dromaeosaurids (Kiernan & Schwimmer, 2004), ornithomimids (Baird & Horner, 1979; Carpenter, 1982; Schwimmer et al., 1993), and ceratopsians (Chinnery et al., 1998; Longrich, 2016; this paper).

Mississippi’s published fragmentary dinosaur remains currently encompass only hadrosaurs (Horner, 1979) and indeterminate theropods (Carpenter, 1982), although one association of over two dozen elements of a single juvenile hadrosaur has been described (Kaye & Russell, 1973). One of the unassigned theropod pedal phalanges (Carpenter, 1982) was later identified as Mississippi’s first known ornithomimid (Baird, 1986). In addition to previously described Mississippi material (Carpenter, 1982), MMNS possesses unpublished, largely isolated elements of hadrosaurs (the most commonly encountered), nodosaurs (teeth and fragmentary bones), dromaeosaurids (teeth), and ornithomimids (the second most common dinosaur). Except for the ceratopsian tooth, all MMNS Mississippi dinosaur holdings (most of it unpublished) are derived from upper Santonian through lower Maastrichtian deposits. Dinosaurs have been reported (Ebersole & King, 2011) but are otherwise undescribed from the upper Maastrichtian of the Gulf Coastal Plain. Many more dinosaur discoveries have been encountered and substantiated in the Maastrichtian of the Atlantic Coastal Plain, namely from the Navesink Formation in New Jersey (see reviews by Weishampel & Young, 1996; Gallagher, 1997).

Conclusion

The ceratopsid tooth from the Owl Creek Formation of Mississippi represents the first unequivocal occurrence of this clade in Appalachia (eastern North America). The fossil is consistent with the hypothesis that clades from Laramidia (western North America) dispersed eastward during the retreat of the Western Interior Seaway sometime during the Maastrichtian. We predict that future work will uncover additional evidence of “western” vertebrate clades in Appalachia; in particular, careful placement within a geological context will help to establish the exact timing and tempo of the seaway retreat.

We extend our gratitude to T. L. Harrell, Jr., who first recognized the tooth as belonging to a ceratopsian and introduced the writers to one another, which led to the current project. Harrell also identified the mosasaur teeth found at the ceratopsian locality (Fig. 3H). Thanks also to P. Kuchirka, MMNS volunteer, who molded/cast the tooth; D. Kitchens, who graciously allowed us access to his property where the tooth was found; M. Garb of Brooklyn College (CUNY), who identified ammonites from the tooth locality, which were useful for biostratigraphic determinations; J. Ebersole of McWane Science Center for assistance with vertebrate fossil identifications; K. Berry for discussion on Cretaceous biogeography; and Colorado Plateau Geosystems for licensed use of the paleogeographic maps. Discussions with F. Varriale were helpful in establishing the orientation of the specimen. Comments from A. Averianov, P. Dodson, D. Fowler, B. McFeeters, H.-D. Sues, and an anonymous reviewer were helpful in revising the manuscript.

Institutional Abbreviations

AZMNH Arizona Museum of Natural History, Mesa, Arizona, USA

MMNS Mississippi Museum of Natural Science, Mississippi Department of Wildlife, Fisheries, and Parks, Jackson, Mississippi, USA

Additional Information and Declarations

Competing Interests

Author Contributions

Data Availability

Andrew A. Farke is an Academic Editor for PeerJ.

Andrew A. Farke conceived and designed the experiments, performed the experiments, analyzed the data, contributed reagents/materials/analysis tools, wrote the paper, prepared figures and/or tables, and reviewed drafts of the paper.

George E. Phillips conceived and designed the experiments, performed the experiments, analyzed the data, contributed reagents/materials/analysis tools, wrote the paper, prepared figures and/or tables, and reviewed drafts of the paper.

The following information was supplied regarding data availability:

MorphoSource, Project P275, Media M10890, http://www.morphosource.org/Detail/SpecimenDetail/Show/specimen_id/4475.

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
