# Peer review of "The first reported ceratopsid dinosaur from eastern North America (Owl Creek Formation, Upper Cretaceous, Mississippi, USA)"

_PeerJ, doi:10.7717/peerj.3342_

## Round 0.1 · original submission · Minor Revisions

This manuscript requires only very minor changes before the Editor can recommend it for acceptance for publication.

·

Basic reporting

no comment.

Experimental design

were no experiments.

Validity of the findings

see the next item.

Additional comments

The paper describes a single ceratpsid tooth, the first record of Ceratopsidae from the eastern North America (Appalachia). The morphology of the tooth, which is well preserved, notable a double root and a prominent, blade-like carina, leaves no doubts that it belongs to a ceratopsid. The authors discuss thoroughly the phylogenetic distribution of the diagnostic characters of the tooth from Mississippi. Although geologic context of the finding is complicating, I agree with authors that reworking from older strata is unlikely due to the state of preservation of the fossil. This is one of the rare opportunity to test the hypothesis that faunal differences between Laramidia and Appalachia were reduced when the two land masses were reunited following the retreat of the interior seaway during the late Maastrichtian. I agree that dispersal of a ceratopsid from Laramida to Appalachia is most likely explanation of this discovery. Appalachia may had some connections with Europe during the Late Cretaceous, as indicated by discovery of a possible zhelestid mammal (Emry et al., 1981), and, possibly, by a multituberculate (Hainina was cited for the latest Campanian Marshalltown Formation of New Jersey, but this material was not described and identification was not confirmed). But ceratopsids currently not known in the Late Cretaceous of Europe. The paper is well organized and provides a wealth of new data. I recommend it for publication in PeerJ as it is.

More specific comments:

Title: “The first reported ceratopsid dinosaur from eastern North America (Owl Creek Formation, Late Cretaceous, Mississippi, USA).” Late Cretaceous is a time when ceratopsid lived, but the formation should be placed in the Upper Cretaceous.

Abstract: “Ceratopsids (“horned dinosaurs”) are known from numerous specimens in western North America and Asia […]” One may read this as were numerous specimens of ceratopsids also in Asia, but they were few.


Emry R.J., Archibald J.D. and Smith C.C. 1981. A mammalian molar from the Late Cretaceous of northern Mississippi. Journal of Paleontology 55: 953-956.

·

Basic reporting

This is an extremely satisfying and satisfactory report. I found it completely convincing, The tooth is precisely what it is claimed to be. It is referred to the correct taxonomic level; taxonomic restrain is observed so that there is no question of applying a binomial to the fossil. The stratigraphic context is expertly described. Likewise the anatomical description is superb. Finally, the biogeographic conclusions are factually based. This MS is so clean that my itchy editorial fingers have been unable to place a single red mark on the text. I congratulate the authors.

Literature Cited excellent!

Experimental design

Entirely satisfactory in design and execution.

Validity of the findings

The conclusions are completely reasonble and follow directly from the data presented.

Additional comments

I will mention just three matters. One is that the find is certainly fortuitous, a veritable pearl in the chaff. Given that the fossil was found in a modern stream lag along with fossils of Pleistocene, Paleocene and Cretaceous age, I find it a bit of a reach to conclude the age of the ceratopsid tooth is “latest Maastrichtian.” This could be taken as meaning “the last 15 minutes of the Mesozoic” or something like that. I appreciate the reasons for thinking it may have come from near the top of the Owl Creek, but given the mixing it may not have as well. Perhaps the less-hyperbolic term “late Maastrichtian” might be advanced as a possibility. Still extremely interesting.

The other thing that I found quite interesting is Fig. 5 detailing withdrawal of the Interior Seaway. We all know that the seaway drained away south to north, but I could not have specified the timing of this event. Do we have a sense of how long the draining took south to north? The fact of a potential southern dispersal corridor to Appalachia is really interesting – an eye-opener.

Could you place the size measurements of the tooth in context? What are dimensions of a Triceratops tooth? Of Centrosaurus/Chasmosaurus tooth. My imopression is that the tooth is of a size consistent with the latter group rather than with Triceratops. A small point but perhaps worth mentioning.

Great job!

Reviewer 3 ·

Basic reporting

No comment.

Experimental design

This is a thorough description of a single ceratopsid tooth.

Validity of the findings

As noted in a comment by Denver Fowler for the PeerJ PrePrint, it would be worth addressing the possibility that this tooth may represent a carcass that had floated from Laramidia.

The tooth is identified as representing a ceratopsid; however, Figure 5 strongly suggests that this tooth more specifically represents a chasmosaurine ceratopsid. It may not be possible to distinguish the teeth of centrosaurines and chasmosaurines, but the stratigraphic and geographic distributions of taxa may offer insights which are explored here only in Figure 5 (though not specifically mentioned in the figure caption) but not in the body of the manuscript. This would be worth addressing or clarifying.

---

## Round 0.2 · accepted · Accept

The revised manuscript is acceptable for publication.